# A Nine-Level Inverter with Adjustable Turn-Off Time for Helicopter Transient Electromagnetic Detection

**DOI:** 10.3390/s23041950

**Published:** 2023-02-09

**Authors:** Fengjiang Peng, Cheng Guo, Zhu Chang, Zilong Yan, Qing Zhao, Xiaoping Huang

**Affiliations:** School of Resources and Environment, University of Electronic Science and Technology of China, Chengdu 611731, China

**Keywords:** inverter, transient electromagnetic detection, adjustable turn-OFF time

## Abstract

The current inverter is the core component of the helicopter transient electromagnetic (HTEM) detection system. It should meet the concerns of low loss, high power, and fast turn-OFF time. This article proposes a new circuit topology based on nine-level inverter technology to overcome the drawbacks of typical PWM (pulse width modulation) inverters, such as switching losses and harmonics. This circuit topology overcomes the shortcomings of the traditional single constant voltage clamp circuit in which the turn-OFF time is not adjustable. Using an inverter with the proposed topology is able to avoid the complex PWM control method and switching loss. In this way, the current rising edge and falling edge of this inverter are also improved effectively. The proposed inverter has adjustable turn-ON-time and turn-OFF time, which is significantly different from the conventional single-clamp inverter. Through subsequent experiments, the inverter proved to have the capability of generating trapezoidal current waveforms. Moreover, by modifying the FPGA (Field Programmable Gate Array) control program, three different turn-OFF times are achieved. The nine-level inverter has a peak current of 1.5 A with an adjustable turn-OFF time from 129 μs to 162 μs. Moreover, the switching frequency of the inverter is reduced from 10 kHz to below 100 Hz. The experimental results further demonstrate that it achieves lower switching losses and more flexible transmission. Our work in this article provides an efficient way to improve the performance of HTEM detection systems.

## 1. Introduction

The transient electromagnetic (TEM) method is an efficient geological detection method in geophysics. The helicopter transient electromagnetic (HTEM) system is widely used in mineral exploration, groundwater surveys, polar exploration, urban construction, tunnel prediction, substation grounding grid diagnosis, and other fields because its detection process is not affected by the geological environment. Generally, the system consists of a bipolar current pulse transmitter, a multi-channel receiver, a transmitting coil, and a low-noise sensing coil [1,2,3,4,5].

HTEM method is a time-domain electromagnetic induction detection method, and its process is divided into three stages: transmission, electromagnetic induction, and receiving data. First of all, the helicopter power supply transmits a bipolar current pulse waveform to the ground through the transmitting coil [6]. At the falling edge of the current pulse, a primary field will be generated, the primary field will propagate downward, and an induced current will be excited when it contacts a benign conductor. The magnitude of the current is related to the conductivity of a good conductor. After the primary field disappears, the induced current excited by a good conductor will not disappear instantly. Still, it will continue to generate a secondary magnetic field, which is received by the receiving coil [7]. Due to the heat loss of the induced current in the well-conductive orebody, the secondary magnetic field decays a time roughly exponentially, forming a transient magnetic field. The secondary magnetic field mainly comes from the induced current in the well-conductive orebody, so it contains geological information related to the ore body. The receiving coil receives the secondary magnetic field, the received data is analyzed and processed, and the orebody and related physical parameters are obtained according to the data [8]. The working principle of the HTEM detection system is shown in Figure 1a [9,10,11,12,13].

Compared to conventional series resonant circuits, using an inverter to generate the transmitting signal has the advantage of achieving flexible control. The high-power current pulse inverter is a critical component of the helicopter transient electromagnetic (HTEM) system. The current pulse inverter should meet engineering indicators such as high power and fast turn-OFF time. The high power will provide the system with a more profound detection distance, and the faster turn-OFF time will provide the system with a better detection effect [14,15,16,17,18,19].

In addition, the trapezoid-like waves are used to transmit current waveforms in the HTEM detection system. The current waveform control is generally divided into rising, stable section control, and falling edge control. The waveform of the rising and stable period should be as stable as possible, and the falling edge should be rapid and linear. And its current amplitude, duty ratio, turn-ON time, and turn-OFF time are determined by the switching time of the transistors in the inverter.

At present, a pulse width modulation (PWM) inverter is widely used in the main loop of the HTEM transmission system to obtain bipolar trapezoidal wave transmitting current [20,21]. The generation of this waveform requires the output of three voltages of different amplitudes. The PWM method converts the input high-amplitude voltage into input voltages of different amplitudes by controlling the width of the pulse signal. When the switching frequency is low, the output voltage value differs greatly from the expected value and produces a poor-quality waveform, which will affect the detection quality. Therefore, a switching frequency of 10 kHz or more is often required, and this will result in high switching losses. The nine-level inverter proposed in this paper is able to output different output voltages through the operating states of power devices and only requires a switching frequency of less than 100 Hz to generate the transmitting current, which overcomes the problem of poor waveform quality at the low switching frequency of conventional PWM. On the other hand, it also overcomes the problem of high switching losses at high switching frequencies. From the device point of view, to meet the high current requirements of the detection system, high-power current pulse inverters often require more transistors in parallel to reduce the maximum current borne by a single transistor. Therefore, when the system has more transistors, the switching losses will increase exponentially if the traditional PWM method is used to generate bipolar trapezoidal currents.

For the technical problems mentioned above, Jilin university proposes a Semi-periodic mirror symmetry selective harmonic elimination pulse width modulation (SHEPWM) based on a subsection control approach to balance transmitting current quality and switching loss [21]. This is a very meaningful study, under the condition of lower switching frequency, if the inverter circuit uses the SHEPWM method to generate the current signal, then compared with the PWM method, the SHEPWM method develops a better quality of the transmitting signal. It is mainly reflected in that the reverse overshoot current is smaller and the fluctuation stability of the flat top is better. Because the basic principle of SHEPWM is to solve the Fourier series of the segmented control voltage and use an iterative calculation to make the Fourier series of the pulse signal generated by the inverter approach, it continuously and then accurately controls the switching time of the switching device (switching angle) to generate the waveform required by the system. This approach achieves high-quality transmission of waveforms with a low switching frequency of 1 kHz. However, this approach is also difficult to solve for the switching angle time, and the initial value is difficult to choose. This nine-level inverter-based approach proposed in this paper avoids the complex solution of the switching angle and can transmit signals at a low switching frequency.

To meet the requirements of the adjustable turn-OFF time of the inverter, Liu et al. introduced a double clamping current inverter to replace the conventional clamping technology with a nonadjustable turn-OFF time [22], which helps generate a better detection effect. By working with double constant voltage clamping technology, the proposed approach can adjust the turn-OFF time of the current of the inverter to control varied clamping voltage [23]. Different from the above approach, the inverter proposed in this paper makes the turn-OFF time of the transmitting current adjustable by changing the control method of the FPGA, which is done by changing the control software program to adjust the turn-OFF time instead of changing the hardware as described above. The way of Control program modifications is easier to implement than the way of hardware modifications.

In this article, a simple nine-level Inverter will be applied in HTEM. Due to the multiple DC levels and its simple control mode, using a nine-level inverter in HTEM has three main advantages compared with the conventional PWM inverters: the higher voltage capability, the flexible control of turn-OFF time, and the lower circuit switching losses. Unlike the traditional PWM method, the designed current inverter has more flexible turn-OFF characteristics and requires fewer switching times. The turn-OFF time is adjustable by changing the transistor control program generated by the FPGA. The nine-level inverter has many highlights when powered by the helicopter generator. The working principles, coil modeling, circuit design and analysis, and the experimental results of the current inverter are described clearly in the following sections.

## 2. Working Phase and Analysis of Current

The HTEM system consists of a bipolar current pulse transmitter, a multi-channel receiver, a transmitting coil, and a low-noise sensing coil. In the transmitting circuit, if the distributed capacitance is ignored, the transmitting coil of the system is equivalent to a series connection of a resistive element and an inductive element. The equivalent model of the transmitting coil is shown in Figure 1b [24].

From the Biot-Savart law, it follows that the relationship between the magnitude of the magnetic induction intensity dB generated by the current element Idl at a point P in space can be characterized as [25].
(1)$$\tilde{B}=\int_{L}\frac{\mu_{0}I}{4\pi }\frac{dl\times \tilde{e_{r}}}{r^{2}}$$

When the radius of the transmitting coil circuit is R_T_, the number of turns of the coil is N_T_ and the magnitude of the current flowing through the coil is I_T_, the magnetic induction intensity B_T_ at its center point generated by the transmitting coil can be written as [25].
(2)$$B_{T}\left(t\right)=\frac{\mu_{0}N_{T}I_{T}}{2R_{T}}$$

Therefore, when the diameter and turns of the transmitting coil are fixed, the key parameters of the compensation coil are derived from the above equation. This method can be used as a reference basis for the design of the parameters of multiple coils of the HTEM system. Generally, the transmitting coil of the HTEM system is usually regarded as an inductive load. The coil capacitance value Cs is often much smaller than the inductance value Ls, so the influence of capacitance can be ignored in the calculating circuit parameters. The transmitting current is usually a bipolar trapezoidal current, so the inverter plays a critical role in controlling the shape and polarity of the current. Figure 2a is the corresponding current waveform in the coil for the positive current stage; the formation of the entire trapezoidal current is divided into four stages. The first stage is the current amplitude rising stage. The current rises exponentially during this phase, and its rising rate is proportional to the voltage, so a relatively high voltage is usually used at this stage. The next stage is a current flat-top stage, in which the current amplitude will slowly rise or maintain a stable amplitude. A steady current or a slowly rising current does not require high voltage to maintain, and a low voltage can meet the needs of this stage. The third stage is the current amplitude dropping stage; this stage is the most critical in the whole process. Generally, the faster turn-OFF time is much more proper for shallow TEM surveys, and in contrast, the slower turn-OFF time is much more satisfactory for the deep TEM survey [26,27,28,29]. Then at this stage, a high-amplitude reverse voltage needs to be provided to the coil.

Moreover, the last stage is the current overshoot damping stage. This stage aims to reduce the current in the coil to zero and prevent reverse overshoot current. Therefore, this can be achieved by limiting the voltage across the coil to zero. The time setting for the four stages mentioned above are t0 to t1, t1 to t2, t2 to t3, and t3 to t4; these four time periods are denoted as T1, T2, T3, and T4. The voltage waveform of the inverter output to the coil load is shown in Figure 2b.

On the other hand, the inductance of the transmitting coil is large as several mH, and the current flowing through the coil cannot change abruptly due to the characteristics of the inductor itself. The inductance voltage U_L_ is related to the derivative of coil current i_L_ concerning time. It can be expressed as a function as:(3)$$U_{L}\left(t\right)=L\frac{di_{L}\left(t\right)}{dt}$$

By integrating, the coil current can be characterized as a piecewise function of voltage and time as follows:(4)$$I_{c}(t)=\left\{\begin{array}{ll} \frac{U_{c_{1}}}{R_{c}}(1-e^{-\frac{R_{c}}{L_{c}}(t-t_{0})}) & t_{0}\leq t<t_{1}\\ \frac{U_{c_{2}}}{R_{c}}(1-e^{-\frac{R_{c}}{L_{c}}(t-t_{1})})+I(t_{1})e^{-\frac{R_{c}}{L_{c}}(t-t_{1})} & t_{1}\leq t<t_{2}\\ \frac{U_{c_{3}}}{R_{c}}(1-e^{-\frac{R_{c}}{L_{c}}(t-t_{2})})+I(t_{2})e^{-\frac{R_{c}}{L_{c}}(t-t_{2})} & t_{2}\leq t<t_{3} \end{array}\right.$$

In the above formula, I(t_1_) and I(t_2_) can be written as:(5)$$\left\{\begin{array}{l} I(t_{1})=\frac{U_{c}{}_{{}_{1}}}{R_{c}}(1-e^{-\frac{R_{c}}{L_{c}}(t_{1}-t_{0})})\\ I(t_{2})=\frac{U_{c}{}_{{}_{2}}}{R_{c}}(1-e^{-\frac{R_{c}}{L_{c}}(t_{2}-t_{1})})+\frac{U_{c}{}_{{}_{1}}}{R_{c}}(1-e^{-\frac{R_{c}}{L_{c}}(t_{1}-t_{0})})e^{-\frac{R_{c}}{L_{c}}(t_{2}-t_{1})} \end{array}\right.$$

The turn-OFF stage of transmitting current directly influences the signal at early time channels [28]. As different switching times and different maximum current amplitudes will result in different detection effects, the variety of transmitting current waveforms will allow for richer detection of information. However, it is hard for the traditional transmitting circuit to generate various kinds of currents which are shown in Figure 3.

## 3. Circuit Design of Nine-Level Current Inverter

Based on the above formula, it can be seen that the current amplitude can be adjusted by changing the voltage and time when the coil parameters remain unchanged, and when the current peak is fixed, higher voltage amplitudes can shorten the time of the process. Therefore, the focus of the transmitting circuit of the HTEM detection system is to provide a flexible supply of voltage.

In general, using the PWM method to change the voltage pulse width is a way to offer a flexible voltage supply. Figure 4a shows the working process of PWM and the generation of voltage pulses and Figure 4b shows the single inverter circuit used by the traditional HTEM transmitting system. Switching frequency is a critical element that affects the quality of transmitting waveform and the accuracy of HTEM detection. However, obtaining a high-quality voltage waveform through PWM requires relatively high switching, which will introduce a lot of unnecessary switching losses. Generally speaking, the switching loss of the power devices increases with increasing switching frequency. On the other hand, the higher switching frequency will place a greater burden on system heat dissipation, and greater heat dissipation will require a heavier bulk device, which will also affect the detection efficiency of the system. Figure 4c is a traditional voltage clamping circuit used in HTEM transmitting, this circuit doesn’t need a high voltage source and it can use its brief high clamping voltage to accelerate the turn-ON time and turn-OFF time of the transmit current, which gives it an advantage over conventional PWM circuits in terms of the current generation. However, the transmitting current of the voltage clamping circuit is difficult to adjust when the parameters of the circuit are fixed. Therefore, it is necessary to propose a transmitting circuit with low switching losses and flexible control characteristics and which is able to transmit the current in Figure 3.

This paper proposes a nine-level inverter with a simple structure to achieve flexible voltage output to solve the above problems. Figure 5 is the proposed nine-level current inverter circuit, which converts the DC voltage to bipolar current pulses. This inverter uses nine SiC MOSFET S1~S9 as a power device instead of IGBT [30,31]. Compared with a traditional power device IGBT, SiC MOSFET can maintain low on-resistance in a broader range. The switching of SiC MOSFET characteristics are excellent, which can handle high power and high-speed switching, and this inverter consists of S1, S2, S3, and S4 of the DC voltage control module, and S5, S6, S7, and S8 of the full-bridge circuit and dummy load. A voltage control module can achieve the output of nine different voltage levels, and full-bridge modules enable the selection of positive and negative voltage polarity. The dummy load prevents the power supply from idle when S9 is turned on.

By adjusting the switch state of S1~S4, four levels of voltage, V1, V1 + V2, V1 + V3, and V1 + V2 + V3, can be provided to the coil load. The selection of the different voltage levels is based on the turn-OFF time. The shorter the turn-OFF time, the higher the voltage level required. The switching states of S5, S8, and S6, S7 determine the current polarity. When S5 and S8 are turned on, the current is positive; the other mode is negative. V1 is different from V2 and V3; V1 must be present in the circuit when the circuit current is not zero. Because the flat top current stage only needs a low voltage to maintain the current amplitude, it is worth noting that V1 must be a common voltage source. Table 1 shows the ON/OFF states of SiC MOSFET under varying voltage levels. This proposed circuit topology produces a nine-level output voltage using 9 SiC MOSFET. The nine-level output voltage switching strategy is shown below Table 1.

The voltage value V1 + V3 is only one of the options; V1, V1 + V2, V1 + V3, and V1 + V2 + V3 can all be used as the power supply of the coil. If we want the T1 to be as short as possible, the voltage value V1 + V2 + V3 is the best choice. Figure 6b shows the circuit of the flat-top current during the T2, and the low voltage DC source V1 provides the voltage of the coil, and the current amplitude keeps stable or rises slowly, so the voltage source selection scheme at this stage is single. It is worth noting that V1 in the circuit must be a low-voltage source; when the voltage across the coil is not zero, V1 is the lowest voltage level of voltage in the selectable range. Figure 6c is the circuit of the falling edge clamping circuit during the T3. A negative high voltage helps reduce the turn-OFF time of the coil. Therefore, at this stage, it is necessary to convert the positive polarity voltage to the negative polarity voltage, and a high voltage must be provided to the coil. However, a power supply with a voltage value of −V1 − V2 − V3 is a good choice for the HTEM detection system to complete complex and diverse geological detection tasks. The working voltage of the coil at this stage can be selected from −V1 − V2, −V1 − V3, and −V1 − V2 − V3. In addition, −V1 is unsuitable for actual detection tasks due to its low voltage resulting in an excessively long turn-OFF time. The circuit conduction modes of the other two different levels of negative polarity voltage are shown in Figure 7. If the same transistor control method is used at times other than T3, and the three different conduction methods in Figure 7 are used at T3, three different turn-OFF times can be generated with the same current amplitude.

Figure 6d is the current overshoot damping circuit during the T4. At this stage, the task of the inverter is to provide zero voltage to the coil. There are various circuit switching modes to provide zero voltage; selecting a state close to the previous state can reduce the number of switches in the circuit.

The same analysis can be applied to the other states, as presented in Figure 8. It shows the whole working process of the nine-level current inverter at negative polarity trapezoidal current pulse. Figure 8a is the circuit of the rising edge clamping circuit during T1. Figure 8b shows the circuit of flat-top current during the T2. Figure 8c is the circuit of the current amplitude dropping stage during the T3. Figure 8d is the current overshoot damping circuit during the T4. In negative operating mode, the circuit conduction state and voltage selection are very similar to positive polarity; the only difference between them is the conduction state of S5, S8, and S6, S7, which determines the current polarity.

MOSFET is not an ideal device, and the device itself has inevitable turn-ON delay, rise time, turn-OFF delay time, and fall time. These non-ideal factors will lead to a short circuit in the power supply. It is also worth paying attention to the fact that in the design of the control program, we need to set some dead time to avoid a power short circuit. Figure 9 is a common way to generate a dead time to protect the circuit; the dead time value should be greater than the sum of the rise time and turn-on delay of the MOSFET. This nine-level inverter needs to establish a dead time for S1 and S2, S3 and S4, S5 and S7, and S5 and S8, respectively, to better protect the circuit.

## 4. Simulation Results

To evaluate the performance of the nine-level inverter, computer simulations of the current waveforms generated by its different circuit control methods were carried out in Matlab/Simscape. Simulation parameters are as follows: input DC voltages V1, V2 and V3 are 1.2 V, 3 V, and 6.5 V, respectively, load resistance 0.12 Ω, load inductance 2 mH, output bipolar current frequency 25 Hz, and the value of the sampling resistor is 0.04 Ω. These parameters used in the simulation experiments above are consistent with the parameters used in the actual experiments.

In simscape, three different control methods are designed to provide three different negative voltage values to the load during T3 time through the m function, to obtain three different current turn-OFF times, Table 2, Table 3 and Table 4 show the switches in the on-state for the three different control methods. To better compare the three different control methods, the control methods are the same in time T1 and T2. In the three methods, the output voltage of the inverter is V1 + V2 during the time T1, and the output voltage of the inverter during the time T2 is V1 so that the resulting current peaks are the same. In the following simulation, T1 is set to 0.5 ms and T2 is set to 2.5 ms under the condition of the same current peak value; T3 (turn-OFF time) is a variable whose magnitude is inversely proportional to the magnitude of the voltage during this period.

The previous three tables show the serial numbers of the MOSFET in the on-state under the three different control methods. Use the three control methods to realize the control of the nine-level inverter; the output voltage of the sampling resistor is shown in Figure 10, and the actual current value is 25 times the captured voltage signal. According to the simulation results, it can be concluded that the peak current in the three methods is about 2.025 A. In method II, the current turn-OFF time is about 238 μs. Since method III has the smallest output voltage during T3, this method has the longest current turn-OFF time of 283 μs. On the contrary, method IV has the shortest current turn-OFF time of 196 μs due to its largest negative output voltage during T3.

From the above simulation results, it can be concluded that the nine-level inverter provides different levels of a negative voltage to the load to achieve different off-time. However, the simulation value of Simscape is closer to the actual value than Simulink due to its detailed adjustable device parameters. However, there will be considerable power supply noise in the virtual circuit. The loop impedance is larger than the simulated circuit so the simulated turn-OFF time will be slightly larger than the actual value.

## 5. Experimental Results

Figure 11 is the experimental system used in this paper. This experiment is to verify the feasibility of the proposed circuit structure in the transient electromagnetic detection system. The experimental parameters are as follows: the input DC voltages V1, V2, and V3, are 1.2 V, 3 V, and 6.5 V, respectively, and the corresponding filter capacitors C1, C2, and C3 are 10 μF. The inductor load resistance R_c_ is 0.12 Ω, and the load inductance L_c_ is 2 mH. Infineon’s SiC MOSFET AIMW120R045M1 was selected as the main switching element due to its superior performance. The generated bipolar trapezoidal current frequency is 25 Hz, and its duty cycle is about 16%.

In practical engineering, the measurement of the current is often tricky. In this experiment, a 0.04 Ω sampling resistor is used to convert the current signal to be measured into a voltage signal to replace the current probe. However, the voltage signal measured in this way is relatively small, and the power supply ripple will negatively impact the measurement.

The output voltage values of the voltage selection module for the three different control methods I, method II, and method III can be observed in Figure 12a–c. According to the oscilloscope reading, the voltage obtained by the load from the inverter is about 7.6 V during T1, about 1.1 V during T2, and about 4 V during T3. To boost the current, 7.6 V is used, and 1.1 V is used to maintain the current when S9 is on. These three control methods have three different output voltages during T3, about 4 V, 7.5 V, and 10.6 V, respectively, and these negative voltages are used to accelerate the decay of the current.

After connecting the sampling resistor and coil load to the inverter circuit, the current value of the load circuit can be deduced from the sampled voltage, and the actual current value is 25 times the captured voltage signal. The voltages across the sampling resistors obtained after executing three different control methods using the FPGA are shown in Figure 13; it can be seen that the peak load current of the circuit is about 1.7 A. As shown in Figure 13a, the overall shape of the current obtained by method I can be reflected, the current turn-OFF time of this method is about 162 μs. Similarly, as shown in Figure 13b,c, the turn-OFF time of method II is about 142 μs, and method III obtains the shortest turn-OFF time of 129 μs.

Figure 14 shows the bipolar trapezoidal voltage signal on the sampling resistor collected by the oscilloscope. It verifies that the proposed circuit can undertake the task of transmitting the current of the transient electromagnetic detection system. The voltage peak value across the sampling resistor is about 58 mV. The current flowing through the load is more than 1.45 A. The experiment’s attenuation ratio of the oscilloscope probe is in the Figure 14 is 10 times.

The above experiments demonstrate that the proposed inverter can output different amplitudes of voltage to achieve different current turn-OFF times by changing the control mode during T3 time while transmitting the same current amplitude. In the traditional PWM method, the voltage amplitude is regulated by the duty cycle, which often requires a switching frequency of more than 10 kHz, whereas the proposed inverter only requires a switching frequency of less than 100 Hz. This article tested the waveforms generated by three different control methods; the current peaks they produce are close, but they have three different turn-OFF times of 162 μs, 142 μs, and 129 μs, respectively. The turn-OFF time obtained by the experimental test is shorter than the simulation results due to the impedance in the actual circuit. Because the voltage collected by the sampling resistance is small, the power ripple influences the experimental results, but the above experiments can conclude the changing trend of load current.

## 6. Conclusions

For the transmit part of the HTEM system, achieving high quality and low loss output current is the critical point. In this study, a nine-level inverter is proposed to undertake the function of waveform generation and achieve an adjustable turn-OFF time in the HTEM system. The circuit design and its working methods are analyzed in the paper and the proposed inverter overcomes the switching loss caused by the traditional PWM transmit method. In addition, this inverter uses the necessary low-voltage source V1 to reduce the number of MOSFETs in the circuit, and it allows for a greater transmitting current by increasing the supply voltage and extending the turn-ON time T1. Compared with the conventional transmitting circuit, the turn-OFF time of the nine-level inverter is adjustable. Experiments show that the nine-stage inverter has a peak current of 1.5 A and has three adjustable turn-OFF times of 162 μs, 142 μs, and 129 μs. The switching frequency of the inverter is reduced from 10 kHz to below 100 Hz. In this paper, we refer to this structure of multilevel inverters and apply the proposed nine-level inverter to HTEM systems as a concrete implementation. More levels of inverters can output more different levels of voltage to make the current waveform with more adjustable turn-OFF time, which will also increase the number of DC power supplies. Based on the theoretical analysis and experimental verifications, the design method in the proposed inverter provides an efficient way to realize transient electromagnetic detection and enhance the performance and flexibility of the HTEM detection system.

## Figures and Tables

**Figure 1 sensors-23-01950-f001:**
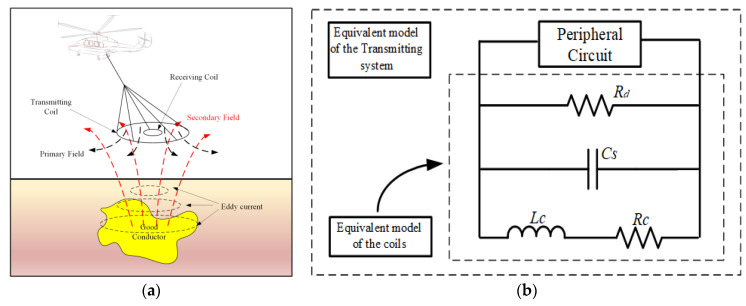
(**a**) Principle of the helicopter transient electromagnetic system. (**b**) Equivalent model of the transmitting system.

**Figure 2 sensors-23-01950-f002:**
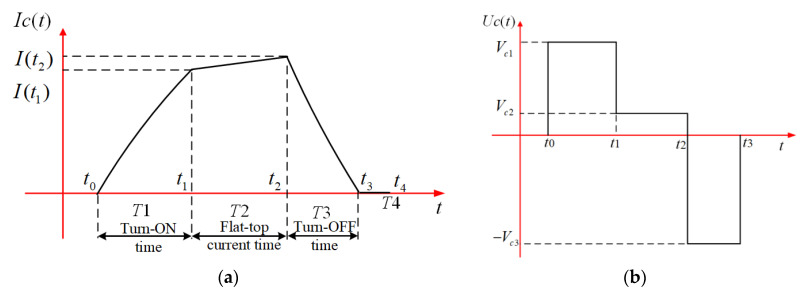
The voltage and current waveform: (**a**) Current excited by the voltage in the coil load (**b**) the output voltage waveform of the inverter.

**Figure 3 sensors-23-01950-f003:**
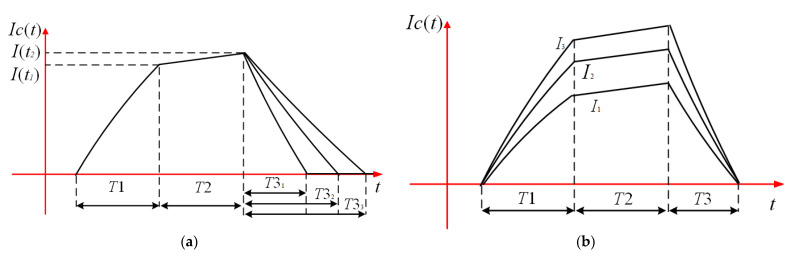
Two types of adjustable current waveform: (**a**) the same maximum amplitude currents with different turn-OFF times, (**b**) the same turn-OFF time currents with different maximum amplitude.

**Figure 4 sensors-23-01950-f004:**
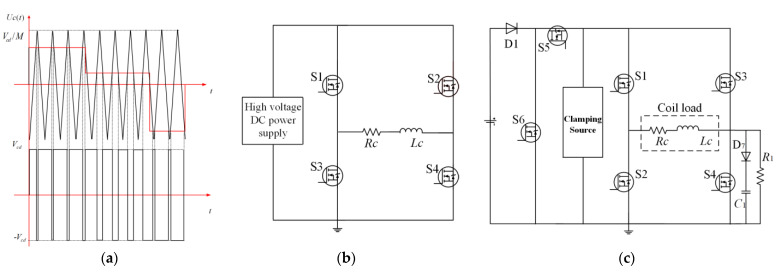
Common HTEM current generation methods and circuits: (**a**) the process of generating PWM voltage pulses, (**b**) the single inverter circuit is used by the HTEM transmitting system, (**c**) the clamping voltage circuit is used by the HTEM transmitting system.

**Figure 5 sensors-23-01950-f005:**
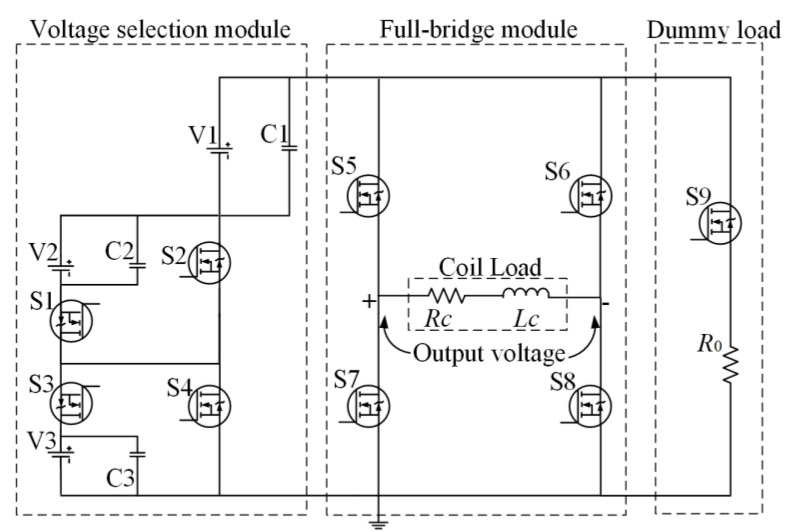
The proposed nine-level current inverter circuit.

**Figure 6 sensors-23-01950-f006:**
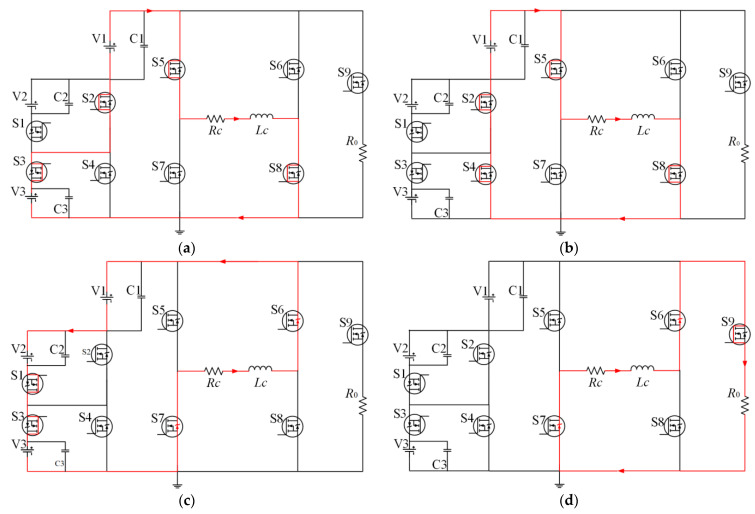
Operating model of the positive pulse current: (**a**) current amplitude rising stage in T1, (**b**) flat-top current stage in T2, (**c**) current amplitude dropping stage in T3, (**d**) current overshoot damping stage in T4.

**Figure 7 sensors-23-01950-f007:**
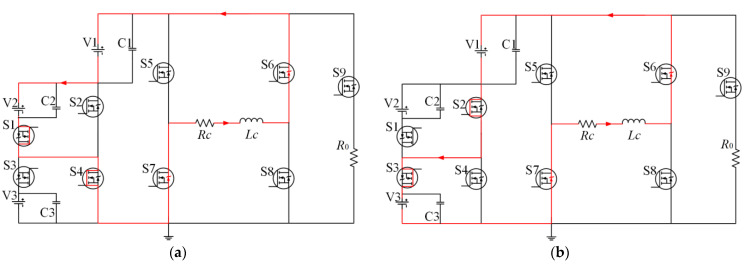
Operating model of the three voltage levels: (**a**) −V1−V2 in T3. (**b**) −V1−V3 in T3.

**Figure 8 sensors-23-01950-f008:**
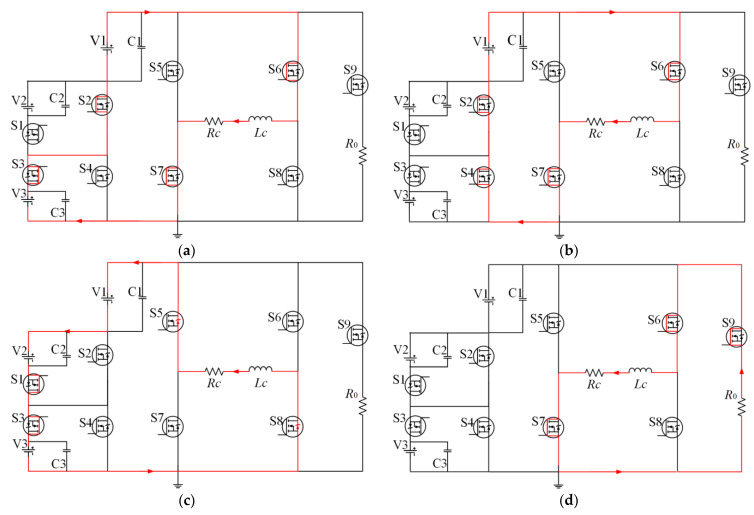
Operating model of the negative pulse current: (**a**) current amplitude rising stage in T1, (**b**) flat-top current stage in T2, (**c**) current amplitude dropping stage in T3, (**d**) current overshoot damping stage in T4.

**Figure 9 sensors-23-01950-f009:**
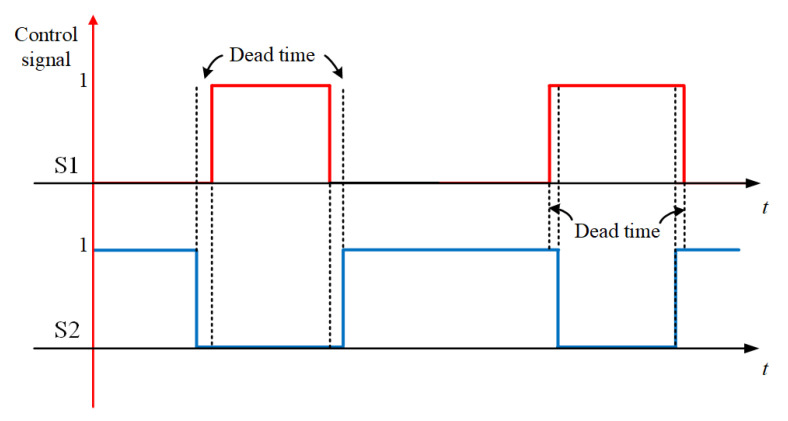
Common ways to generate pulse dead time.

**Figure 10 sensors-23-01950-f010:**
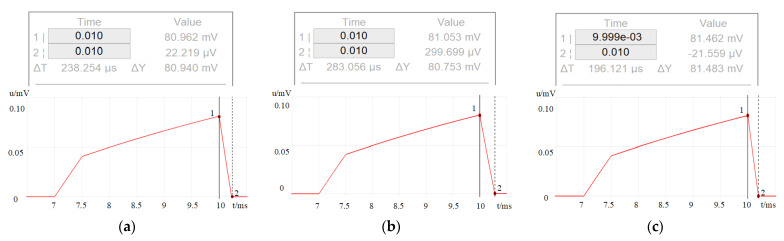
The output voltage of the sampling resistor from different control methods: (**a**) in control method I, the output voltage across the sampling resistor, (**b**) in control method II, the output voltage is across the sampling resistor, (**c**) in control method II, the output voltage is across the sampling resistor.

**Figure 11 sensors-23-01950-f011:**
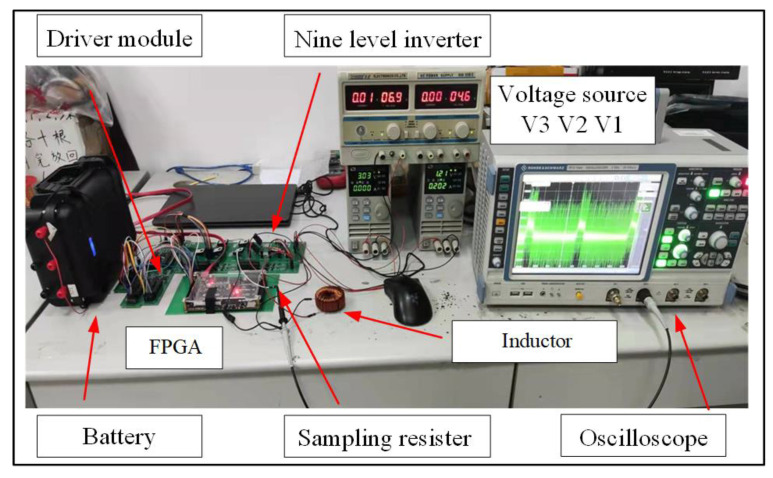
Experimental setup.

**Figure 12 sensors-23-01950-f012:**
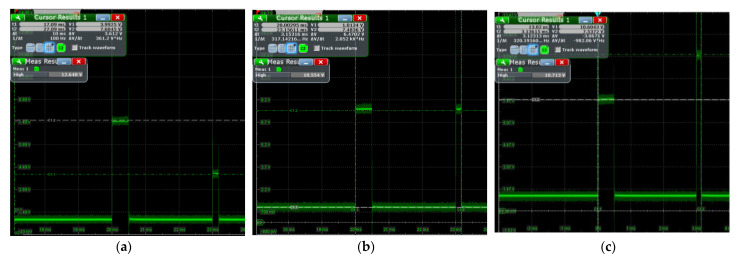
The output voltage of the voltage selection module at no load: (**a**) the output voltage in method I, (**b**) the output voltage in method II, (**c**) the output voltage in method III.

**Figure 13 sensors-23-01950-f013:**
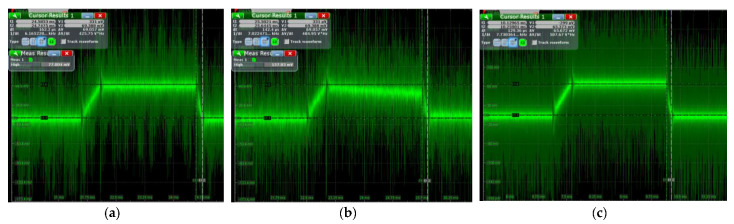
The output voltage of the sampling resistor: (**a**) the sampling voltage obtained from method I, (**b**) the sampling voltage obtained from method II, (**c**) the sampling voltage was obtained from method III.

**Figure 14 sensors-23-01950-f014:**
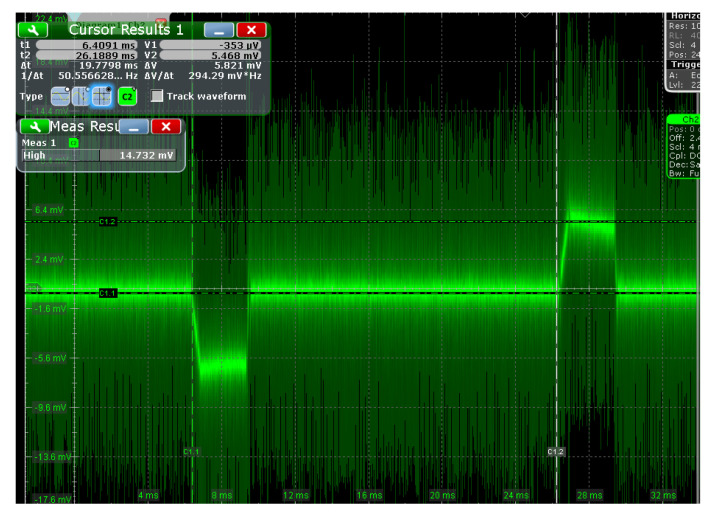
Bipolar trapezoidal voltage signal on the sampling resistor.

**Table 1 sensors-23-01950-t001:** Output voltage according to the switches on conditions.

Output Voltage	Switches in the On-State			
V1	S2	S4	S5	S8
V1 + V2	S1	S4	S5	S8
V1 + V3	S2	S3	S5	S8
V1 + V2 + V3	S1	S3	S5	S8
−V1	S2	S4	S6	S7
−V1 − V2	S1	S4	S6	S7
−V1 − V3	S2	S3	S6	S7
−V1 − V2 − V3	S1	S3	S6	S7
0	S2 S4 S5 S6 S9			

**Table 2 sensors-23-01950-t002:** Circuit control method I.

Output Voltage	Time	Switches in the On-State			
V1 + V3	T1	S2	S3	S5	S8
V1	T2	S2	S4	S5	S8
− V1 − V3	T3	S1		S4	

**Table 3 sensors-23-01950-t003:** Circuit control method II.

Output Voltage	Time	Switches in the On-State			
V1 + V3	T1	S2	S3	S5	S8
V1	T2	S2	S4	S5	S8
− V1 − V3	T3	S2		S3	

**Table 4 sensors-23-01950-t004:** Circuit control method III.

Output Voltage	Time	Switches in the On-State			
V1 + V3	T1	S2	S3	S5	S8
V1	T2	S2	S4	S5	S8
−V1 − V2 − V3	T3	S1		S3	

## Data Availability

Not applicable.

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
