# Peer review of "A Nine-Level Inverter with Adjustable Turn-Off Time for Helicopter Transient Electromagnetic Detection"

_sensors, 2023, doi:10.3390/s23041950_

Round 1

Reviewer 1 Report

Interesting work and topic. To improve its quality, a few concerns have to be addressed:

-Some equation must be referenced;

-Is it possible to increase the number of levels further? what would be the point?

Reviewer 2 Report

 This work presents ‘A nine-level inverter with adjustable turn-OFF time for heli-2 copter transient electromagnetic detection’ in an interesting manner. However, the following comments should be considered for a detailed technical improvement of the work.

 The paper requires further English revision. 

The novelty of the work is not clear the authors should clearly Highlight the novel/contribution of your work.

Some parameters are  not defined.

Figure 5.b does not exist.

Check that the figures and block diagrams are of good quality. 

-“This article proposes a new circuit topology based on nine-level inverter technology to overcome the drawbacks of typical PWM inverters, such as switching losses and harmonics and the fixed  turn-OFF time of a traditional single constant voltage clamping circuit”. it is recommended to add a comparative study to show the mentioned advantages of the proposed inverter technology.
